# 2-Bromopalmitate decreases spinal inflammation and attenuates oxaliplatin-induced neuropathic pain via reducing Drp1-mediated mitochondrial dysfunction

Zhi-Bin Dong[1]☉, Yu-Jia Wang[1]☉, Meng-Lin Cheng[2]☉, Bo-Jun Wang[2], Hong Lu[2], Hai-Li Zhu[2], Ling Liu[2]*, Min Xie[2]*

1 School of Pharmacy, Hubei University of Science and Technology, Xianning, Hubei, China, 2 School of Basic Medical Sciences, Hubei University of Science and Technology, Xianning, Hubei, China

☉ These authors contributed equally to this work.
* liuling0306@163.com (LL); xiemin2018@hbust.edu.cn (MX)

**Data Availability Statement:** All relevant data are within the paper.

## Abstract

Oxaliplatin (OXA) is a third-generation platinum compound with clinical activity in multiple solid tumors. Due to the repetition of chemotherapy cycle, OXA-induced chronic neuropathy presenting as paresthesia and pain. This study explored the neuropathy of chemotherapy pain and investigated the analgesic effect of 2-bromopalmitate (2-BP) on the pain behavior of OXA-induced rats. The chemotherapy pain rat model was established by the five consecutive administration of OXA (intraperitoneal, 4 mg/kg). After the establishment of OXA-induced rats, the pain behavior test, inflammatory signal analysis and mitochondrial function measurement were conducted. OXA-induced rats exhibited mechanical allodynia and spinal inflammatory infiltration. Our fluorescence and western blot analysis revealed spinal astrocytes were activated in OXA rats with up-regulation of astrocytic markers. In addition, NOD-, LRR- and pyrin domain-containing 3 (NLRP3) inflammasome mediated inflammatory signal cascade was also activated. Inflammation was triggered by dysfunctional mitochondria which represented by increase in cyclooxygenase-2 (COX-2) level and manganese superoxide dismutase (Mn-SOD) activity. Intrathecally injection of 2-BP significantly attenuated dynamin-related protein 1 (Drp1) mediated mitochondrial fission, recovered mitochondrial function, suppressed NLRP3 inflammasome cascade, and consequently decreased mechanical pain sensitivity. For cell research, 2-BP treatment significantly reversed tumor necrosis factor-α (TNF-α) induced mitochondria membrane potential deficiency and high reactive oxygen species (ROS) level. These findings indicate 2-BP decreases spinal inflammation and relieves OXA-induced neuropathic pain via reducing Drp1-mediated mitochondrial dysfunction.

## Introduction

According the statistics of International Agency for Research on Cancer, there will have 4.82 million new cancer cases, and almost 3.21 million cancer deaths in China, in 2022 [1].

**Funding:** This study was supported by the grants from the National Natural Science Foundation of China (Nos. 81971066, 81901149 and 32100823), Research Project of Hubei Provincial Department of Education (Nos. Q20192807, B2019167), Hubei University of Science and Technology Program (Nos. 2020TD02, BK202116). The funders had no role in the study design, data collection and analysis, decision to publish.

**Competing interests:** The authors have declared that no competing interests exist.

Chemotherapy is the important therapeutic approach in cancer, which induces a commonly adverse effect named chemotherapy-induced peripheral neuropathy [2]. Oxaliplatin (OXA) is a third-generation platinum compound with clinical activity in multiple solid tumors [3]. OXA-induced neuropathy is characterized by an acute neurotoxicity and a chronic cumulative neuropathy [4]. Acute neurotoxicity occurs in the hours and days following OXA infusion. Chronic neuropathy in particular with axonal neuropathy presenting as paresthesia and pain, is usually happened due to the repetition of chemotherapy cycle, and lasts for several years after chemotherapy ends [5]. According to statistics, 30–50% of patients suffer from OXA-induced chronic neuropathy and affect the quality of life seriously [6]. However, up to now, there is no effective preventive treatment for OXA-induced neuropathy. Therefore, understanding the pathogenesis of OXA-induced neuropathy and developing relative analgesic drugs are urgently needed.

As an anti-cancer agent, OXA induced histopathological changes in the heart, liver, intestines and muscle in a dose dependent manner [7]. Moreover, OXA is known to cause apoptosis and induce the peculiar alterations of nervous tissue, especially axonal degeneration and focal demyelination in dorsal root ganglia [8]. It is reported that OXA induced an increase in firing rates of wide dynamic range neurons, a dramatic effect on oxidative damage, and activation of glial cells in spinal cord [9]. Glial cells play a crucial role in chemotherapy-induced peripheral neuropathy. Activated glial cells (including microglia and astrocytes) accompanied by induction of reactive gliosis, release of pro-inflammatory cytokines, generation of free radicals and occurrence of cytotoxic excitatory amino acids circulation which increases pain hypersensitivity and participates in OXA-induced neuropathy pathogenesis [10–12]. Oxidative damage is harmful for cells and tissues by declining the physiologic function. In particularly, mitochondria are closely linked to oxidative damage [13]. Mitochondrial dysfunction has been implicated as a major patho-mechanism for OXA-induced peripheral neuropathy [9].

The principal functions of mitochondria are ATP production, reactive oxygen species (ROS) generation and maintenance of $Ca^{2+}$ homeostasis [14]. Mitochondria undergo fission and fusion to maintain their function, coupling with high metabolic requirements. Fragmentated and discontinuous mitochondria are patterns of mitochondrial quality control under cellular stress [15–17]. Fragmented mitochondria are associated with decreased ATP production and decoupling. Mitochondrial dysfunction with impairing homeostasis and increasing ROS production are related to pain [18, 19]. Mitochondrial functions are regulated by the certain mitochondrial proteins which undergo post-translational modifications, such as phosphorylation, ubiquitination, and S-palmitoylation [20]. Modifications of these proteins impacts mitochondrial dynamics and efficiency to generate ATP. ZDHHC13 is a member of the palmitoyl transferase family for catalyzing protein S-palmitoylation, which attached a palmitate group to a specific cysteine residue, enhanced protein hydrophobicity, affected protein membrane association, altered protein subcellular localization and is reported to be an important regulator of mitochondrial activity and dynamics [21]. ZDHHC13 deficiency resulted in decreased level of dynamin-related protein 1 (Drp1) S-palmitoylation accompanies changes in mitochondrial dynamics, increase in glycolysis, glutaminolysis and lactic acidosis, and imbalance of neurotransmitters. Drp1 is a cyto-GTPase, translocation from cytoplasm to mitochondrial outmembrane promoting mitochondrial fission [22]. ZDHHC-13-mediated Drp1 S-palmitoylation enables the occurrence of the fission-fusion process *in vitro* and *in vivo*. 2-bromopalmitate (2-BP) is an irreversible pan-inhibitor of palmitoyl transferase, and has been widely used as a protein palmitoylation inhibitor [23–25]. In the present study, we proposed that 2-BP reduced Drp1-mediated mitochondrial fission and decreased inflammatory response, and consequently relieved OXA-induced neuropathic pain.

## Materials and methods

### Antibodies and reagents

Primary antibodies GFAP rabbit mAb (A19058), Complement Factor B rabbit pAb (A1706), Complement C3 rabbit pAb (A13283), β-actin Rabbit mAb (AC026), NLRP3 rabbit pAb (A12694), Caspase-1 rabbit pAb (A0964), IL-1β rabbit pAb (A1112), Drp1 rabbit pAb (A2586), phosphor-Drp1-S616 rabbit pAb (AP0849), COX2 rabbit mAb (A3560) were purchased from ABclonal Technology (Wuhan, CHN). The secondary antibodies used for Western blotting was HRP Goat Anti-Rabbit IgG (H+L) (AS014) which purchased from ABclonal Technology (Wuhan, CHN). The secondary antibodies used for immunofluorescence analysis was Goat Anti-Rabbit IgG H&L (FITC) (ab6717) purchased from abcam (Cambridge, UK). H&E staining solution (BL735B) was purchased from Biosharp Life Sciences (Hefei, CHN). The Mn-SOD Assay Kit (S0103), Reactive Oxygen Species Assay Kit (S0033S) and Mito-Tracker Red CMXRos (C1049B) were obtained from Beyotime Biotechnology (Shanghai, CHN). OXA (61825-94-3) was obtained from Aladdin Biochemical Technology (Shanghai, CHN). 2-bromopalmitate (2-BP, 18263-25-7) was purchased from Sigma-Aldrich (St Louis, USA).

### Experimental animals

Sprague-Dawley (SD) rats weighing 180–200 g were purchased from Hubei Province Experimental Animal Center (Wuhan, China). All animals were house with *ad libitum* access to water and food in a 12/12 h light-dark cycle regime and controlled temperature room ($22 \pm 1$°C). This study has been approved by the Ethics Committee of Hubei University of Science and Technology (2020-01-900). All experimental procedures in this study were complied with the local and international guidelines on ethical use of animals and all efforts were made to minimize the number of animals used and their sufferings.

### Cell culture and treatment

C6 glial cells (Jennio Biotech, CHN) were cultured in DMEM medium (Gibco, NY, USA) with 10% fetal bovine serum, 50 U/ml penicillin and 50 μg/ml streptomycin (Gibco, NY, USA) at 37°C with 5% carbon dioxide. TNF-α was diluted with 0.9% NaCl. 2-BP was dissolved in dimethyl sulfoxide (DMSO, Beyotime, Shanghai, CHN). C6 cells were seeded on plate and induced by 5 ng/μl TNF-α for 4 h. After inducement, cells were treated with 0 and 1 μM 2-BP for 24 h, then digested with trypsin (Gibco, NY, USA) and collected protein samples for western blot.

### OXA administration and 2-BP treatment

OXA was dissolved in 5% glucose solution and administrated intraperitoneally (i.p.) at 4 mg/kg into each rat for five consecutive days. The same volume of 5% glucose solution was injected to control group [26]. For intrathecal injection, rats were held firmly, and 25-μl microsyringe was inserted between L5 and L6 vertebrae. A sudden advancement of the needle accompanied by a slight flick of the tail was used as the indicator for the proper insertion into the subarachnoid space [27]. 2-BP was injected slowly in a 10 μl volume, and the concentration for rats was 1 mg/kg. 2-BP was dissolved in DMSO and diluted by 0.9% NaCl before use. Vehicle was DMSO and 0.9% NaCl and injected with same volume.

### Mechanical allodynia

Rat was placed on the floor of a $5 \times 5$ mm wire mesh grid and allowed to acclimate quietly for 30 minutes. Von Frey filaments (0.4 g to 26 g, Stoelting, IL, USA) were used to apply

mechanical stimulation to the hind paws. The calibrated monofilaments were applied perpendicularly to the plantar surfaces until the filaments were bent, and a brisk withdrawal was considered as positive response. Whenever a positive response occurred, the von Frey filament with the next lower force was applied, and whenever a negative response occurred, the filament with the next higher force was applied. The pattern of positive and negative withdrawal response was converted to 50% paw withdrawal threshold (PWT) [28].

## H&E staining and immunofluorescence analysis

Hematoxylin and Eosin (H&E) staining was conducted after 7 days of intraperitoneal injection of OXA. Briefly, both control and OXA group rats were deeply anesthetized and transcardiac perfusion with saline followed by 4% paraformaldehyde (PFA) were used to clear blood and preserve spinal cord for H&E staining and immunostaining. Spinal cord tissue was removed, fixed in 10% neutral buffered formalin, embedded in paraffin, and sliced to 4 μm thick (Leica RM2165). Slides were stained using standard H&E methods. Briefly, the slices were treated with xylene, 100% ethanol, 90% ethanol, 70% ethanol for dewaxing and rinsing with tap water for 2 min. Then the slices were dyed with hematoxylin solution for 3 min and stained with eosin solution for 3 min. The dehydration and transparent treatment were conducted by putting the slices into 80% ethanol, 90% ethanol, 95% ethanol, 100% ethanol, xylene. Finally, the slices were sealed with neutral balsam and observed using a microscope (OlympusIX73, Tokyo, Japan) and analyzed by using the ImageJ software.

For immunofluorescence analysis, after dewaxing and antigen retrieval, the slices were incubated with 3% hydrogen peroxide for 10 min and blocked with goat serum at room temperature for 1 h. Then the slices incubated with specific primary antibodies overnight at 4°C. The following primary antibodies were used: anti-GFAP (1:100), anti-NLRP3 (1:100), anti-Drp1 (1:100) and anti-COX-2 (1:100). Subsequently, the slices were incubated with and fluorescent secondary antibody at room temperature for 1 h. Images were taken with fluorescence microscope (Olympus IX73). The fluorescence intensities were analyzed using ImageJ software.

## Western blot analysis

Western blot analysis was performed as previously described [29]. After 2-BP treatment for 12 h, the rats were anesthetized and dislocated to death. Spinal cords of rats from different groups were isolated and washed twice with ice-cold PBS (pH7.4). Then the tissues were homogenized in ice-cold RIPA lysis buffer [50 mM Tris (pH 7.4), 150 mM NaCl, 1% Triton X-100, 1% sodium deoxycholate and 0.1% SDS] containing protease inhibitor mixture (Sigma, St Louis, USA). The homogenates were centrifuged at $12,000 \times$ g per 15 min at 4°C and the supernatants were collected for western blot analysis. Protein concentrations were determined by BCA Protein Assay kit (Beyotime, China). The samples were heated to 95°C for 10 min for the denaturation. Equal amounts of proteins (40 μg) for each sample were loaded and separated by SDS-PAGE gels (10%) for 2 h at 90 V. Then the proteins were transferred to polyvinylidene fluoride (PVDF) membranes (0.2 μm, Merck Millipore, MA, USA) in a Tris-glycine transfer buffer at 275 mA for 1 h. Membranes were subsequently blocked with QuickBlock Blocking Buffer for Western Blot (Beyotime, Shanghai, CHN) at room temperature for 90 min. After blocking, the membranes were incubated with primary antibodies at 4°C overnight, antibody against membrane: GFAP (1:1000), CFB (1:1000), Complement C3 (1:1000), NLRP3 (1:1000), caspase-1 (1:1000), IL-1β (1:1000), Drp1 (1:1000), pSer616-Drp1 (1:500), COX2 (1:1000), and the proteins expression was normalized to the expression of β-actin (1:50000). After incubation of primary antibody, the membranes were washed three times with TBS-T and incubated

with horseradish peroxidase (HRP) Goat anti-Rabbit IgG (H+L) (diluted at 1:10000) for 1 h at room temperature. Finally, the bands were detected using ECL detection reagent and visualized with the iBright 1500 (Invitrogen, CA, USA). The protein expressions of immunoreactive bands were analyzed by ImageJ software.

### ELISA assay

The detection method was according to the previous description [30]. Spinal cord was homogenized and lyzed in ice-cold PBS buffer pH7.4 containing a protease inhibitor cocktail. Then the homogenate was centrifuged at 4˚C and the supernatant was collected for ELISA analysis. The Mn-SOD activity was detected by the Mn-SOD Assay Kit. Briefly, after the preparation of samples, the protein concentration was determined by the BCA assay kit (Beyotime Biotechnology, Shanghai, CHN). The Cu/Zn-SOD inhibitor A and sample or standard were mixed in a 96-well plate and incubated at 37˚C for 1 hour. Then the Cu/Zn-SOD inhibitor B was added and incubated at 37˚C for another 15 min. The reaction working solution was added and mixed thoroughly, incubate at 37˚C for 30 min. Determination of absorbance at 450 nm and the enzyme activity was calculated.

### Assessment of mitochondria membrane potential

Changes in cellular mitochondrial membrane potential was detected using MitoTracker Red CMXRos. After TNF-α inducement and 2-BP treatment, the cell culture medium was removed and the Mito-Tracker Red CMXRos working solution was added, and incubated at 37˚C for 20 min. After the incubation, the Mito-Tracker Red CMXRos working solution was removed and the fresh cell culture medium was added. The cells were observed by fluorescence microscope (Olympus IX73; Olympus, Tokyo, Japan). The fluorescence intensity was analyzed by ImageJ software.

### Assessment of ROS

After TNF-α inducement and 2-BP treatment, changes in ROS content of C6 cells from different group were detected by Reactive Oxygen Species Assay Kit uses fluorescent probe DCFH-DA (2,7-Dichlorodi-hydrofluorescein diacetate). The cell culture medium was removed and diluted DCFH-DA was added. Cells were incubated at 37˚C for 30 min. After incubation, cells were washed three times with serum-free medium. Then the cells were observed under a fluorescence microscope (Olympus IX73). The fluorescence intensities were analyzed by ImageJ software.

### Molecular docking modeling assay

X-ray crystal structure of ZDHHC13 was obtained from the Protein data bank (UniProt ID: Q8IUH4, https://www.uniprot.org/uniprotkb/Q8IUH4/entry). The structure of 2-BP was download from PubChem database (Compound CID: 82145, https://pubchem.ncbi.nlm.nih.gov/#query = 2-bromopalmitate), optimized using ChemBio3D software. Auto Dock vina was used to dock conformation between ZDHHC13 and 2-BP. PyMOL was used to visualize the conformation.

### Statistical analysis

All statistical analyses were conducted with SPSS 21.0 software package and all data were evaluated by one-way analysis of variance (one-way ANOVA) with repeated measures followed by Bonferroni post hoc tests and presented as mean ± SD. Data of PWT analysis were presented as mean ± SEM. Significance was described as $P < 0.05$.

## Results

### 2-BP alleviates OXA-induced mechanical allodynia

Auto Dock data showed that 2-BP inhibited ZDHHC13 activity through binding to the DHHC domain which was required for palmitoyl transferase activity (Fig 1A and 1B). Therefore, we used 2-BP, an irreversible pan-inhibitor of palmitoyl transferase to treat OXA rats. OXA was intraperitoneally injected for 5 consecutive days to induce neuropathy. After administration, mechanical pain sensitivity was detected and PWT values. In comparison with the rats in control group, PWT value of OXA rats on day 7 (after the first OXA injection) was dramatically declined (Fig 1C). The PWT values of control and OXA groups were 16.8 ± 1.5 g and

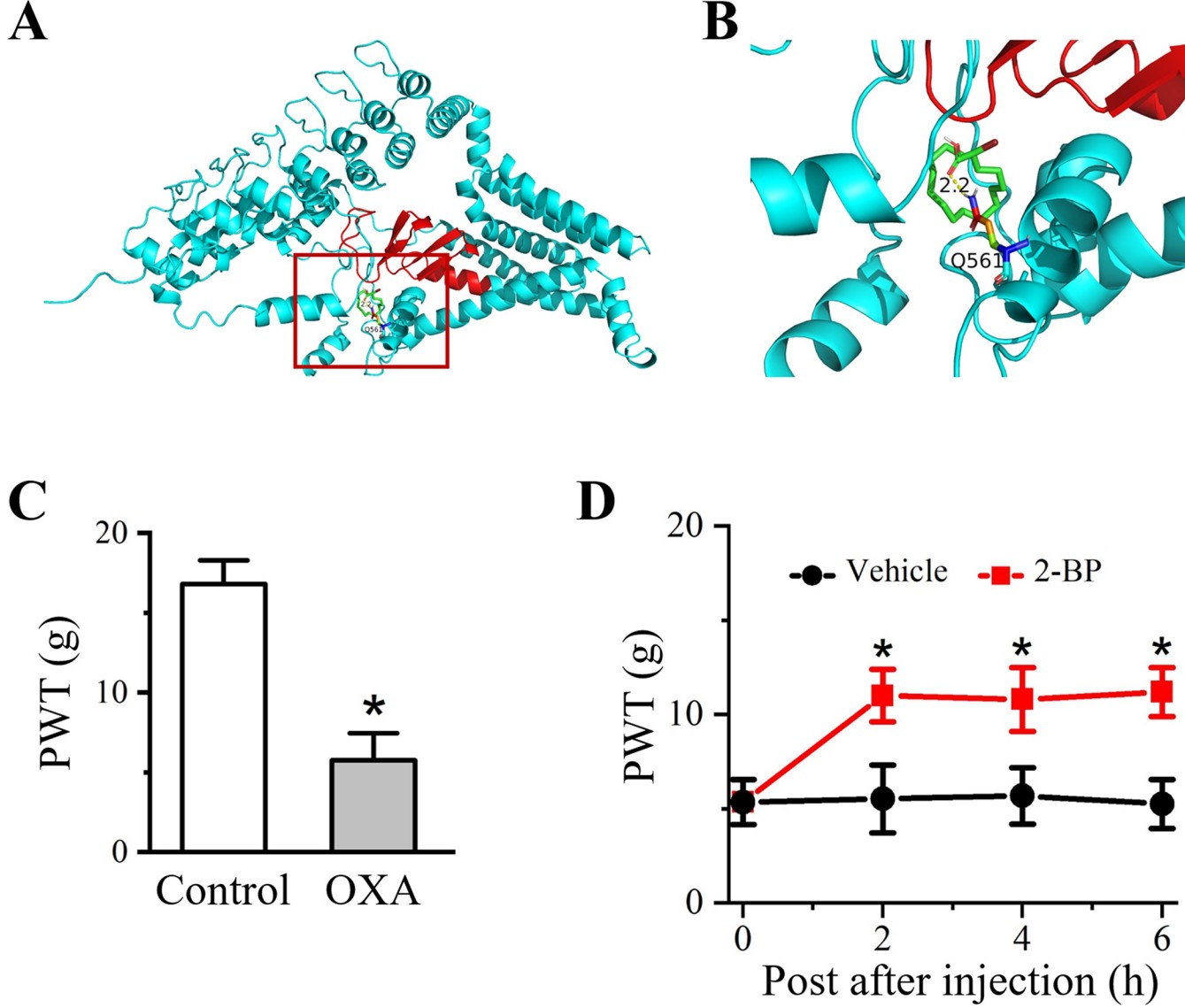

**Fig 1. Detection of pain behavior of rats before and after 2-BP treatment.** (A, B) ZDHHC13-2-BP docking with Autodock. Red rectangular frame showed the DHHC activity domain of ZDHHC13. (C) PWT value of Control and OXA rats at seven days after OXA inducement. Data were expressed as mean ± SEM (n = 6). *$P < 0.05$ compared with the Control group. (D) Changes in PWT value on OXA rats under vehicle and 2-BP treatment. Data are presented as the mean ± SEM (n = 6). *$P < 0.05$ *vs.* OXA group.

5.8 ± 1.7 g ($P < 0.05$ *vs.* control group). The data indicated that consecutively injection of OXA induced mechanical hypersensitivity. To confirm the effect of 2-BP on OXA-induced pain behavior, PWT values were measured after 2-BP administration 2 h, 4 h, and 6 h. As shown in Fig 1D, after 2-BP treatment, mechanical pain sensitivity of OXA rats were significantly decreased, presenting by increased PWT values in 2-BP-treated OXA rats. PWT values of control and OXA group at 2 h, 4 h, and 6 h after 2-BP treatment were 5.5 ± 1.8 g *vs.* 11.0 ± 1.4 g ($P < 0.05$), 5.7 ± 1.5 g *vs.* 10.8 ± 1.7 g ($P < 0.05$), 5.3 ± 1.3 g *vs.* 11.2 ± 1.3 g ($P < 0.05$), respectively. These data suggested that 2-BP had a pain relief effect on OXA-treated rats.

## 2-BP inhibits OXA-induced spinal inflammation and activation of astrocytes

After determination of altered pain behavior in OXA-treated rats upon 2-BP treatment, we subsequently examined the inflammatory infiltration and activation of glial cells to represent the inflammatory response in spinal dorsal horn. H&E-stained sections showed severe infiltration of inflammatory cells in spinal dorsal horn induced by OXA treatment (Fig 2A). Compared with control group, the inflammation score was increased to 1.92 ± 0.18 in OXA group ($P < 0.05$ *vs.* control group, Fig 2B). Glial cell, especially astrocyte, is essential for induction and maintenance of pain, and closely related to inflammatory response [31]. Localization and distribution of glial fibrillary acidic protein (GFAP, astrocytic marker) in spinal cord was detected by immunofluorescence staining. The fluorescence intensity of spinal GFAP was significantly increased upon OXA treatment (Fig 2C). Relative fluorescence intensity of GFAP in OXA-induced rats were 1.48 ± 0.08 ($P < 0.05$ *vs.* control group, Fig 2D). Besides GFAP, we further detected other glial markers to characterize the activated state of astrocytes. A1 astrocyte, insufficiency in normal functions, produced complement components and released toxic factors [32]. This type of astrocyte can be presented by complement factor B (CFB) and complement C3 (C3). Western blot data showed that GAFP, CFB and C3 expression were all increased in spinal cord of OXA rats (Fig 2E). Relative gray values of these proteins in OXA group were 2.87 ± 0.18, 1.67 ± 0.12, and 2.19 ± 0.17, respectively ($P < 0.05$ *vs.* control group, Fig 2F). The activation of astrocytes in spinal cord can be down-regulated by 2-BP treatment as shown in Fig 2E. The relative values of GAFP, CFB and C3 in OXA+2-BP group were decreased to 0.88 ± 0.12, 0.98 ± 0.14, 1.29 ± 0.22 and respectively ($P < 0.05$ *vs.* OXA group, Fig 2F). These data indicated that 2-BP had an inhibitory effect on OXA-induced inflammatory response and spinal astrocytic activation.

## 2-BP suppresses NLRP3 inflammasome mediated spinal inflammation

NOD-, LRR- and pyrin domain-containing 3 (NLRP3) inflammasome plays a critical role in the neuroinflammatory response through mediating caspase-1 activation and pro-inflammatory cytokines secretion [33]. As shown in Fig 3A, the mean intensity of spinal NLRP3 was enhanced in OXA rats and relative intensity value of NLRP3 was 1.76 ± 0.09 ($P < 0.05$ *vs.* control group, Fig 3B). Subsequently, the components of NLRP3 inflammasome cascade were determined by Western blot. The results showed that levels of NLRP3, cleaved-caspase-1 and IL-1β were all up-regulated in the spinal cord of OXA rats (Fig 3C). Relative level of proteins (OXA group) mentioned above were 1.88 ± 0.13, 1.79 ± 0.11 and 1.73 ± 0.13, respectively ($P < 0.05$ *vs.* control group, Fig 3D). These data illustrated OXA administration triggered NLRP3-mediated inflammatory response. The effect of 2-BP on spinal NLRP3 inflammasome cascade was also analyzed. As shown in Fig 3C, 2-BP treatment inhibited activity of NLRP3 inflammasome cascade observed by the decreased expression of NLRP3, cleaved-caspase-1 and IL-1β upon 2-BP treatment. The relative levels of these proteins in OXA + 2-BP group

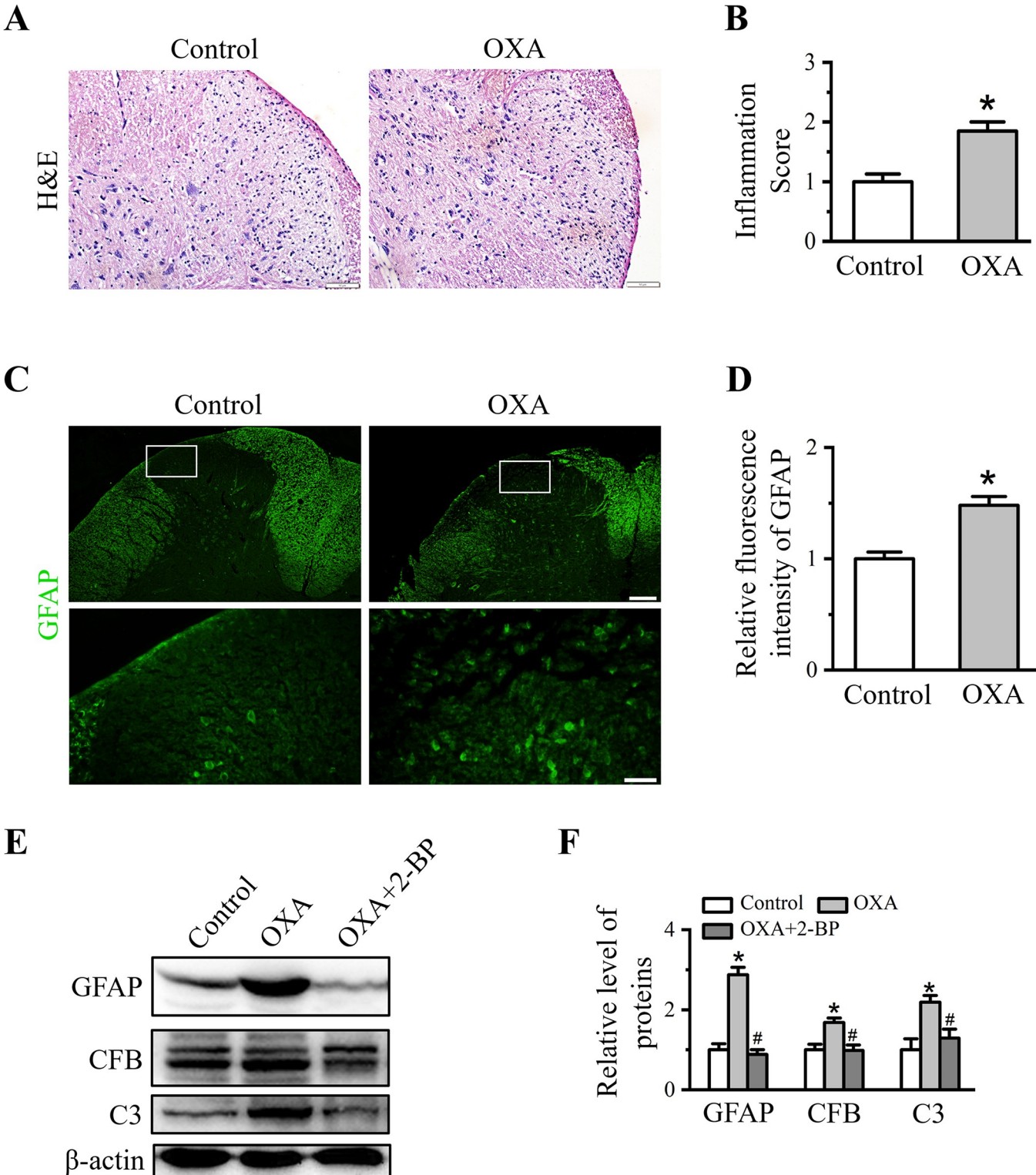

**Fig 2. Effect of 2-BP on the spinal inflammatory infiltration and activation of astrocytes induced by OXA administration.** (A) Representative H&E staining images on the spinal cord of Control and OXA rats. Scale bars: 50 μm. (B) Quantitative analysis of H&E staining. Data were showed as the mean ± SD (n = 3). $^{*}P < 0.05$ *vs.* Control group. (C) Representative immunofluorescence staining of spinal GFAP in Control and OXA groups. Scale bars: first line, 100 μm; second line, 20 μm. (D) Quantitative analysis of the fluorescence intensity of GFAP. Data were showed as the mean ± SD (n = 3). $^{*}P < 0.05$ *vs.* Control group. (E) Western blot analysis of GFAP, CFB and C3 in Control, OXA and OXA+2-BP groups. (F) Quantitative analysis of protein levels in (E). Data were presented as mean ± SD (n = 3). $^{*}P < 0.05$ *vs.* Control group, $^{#}P < 0.05$ *vs.* OXA group.

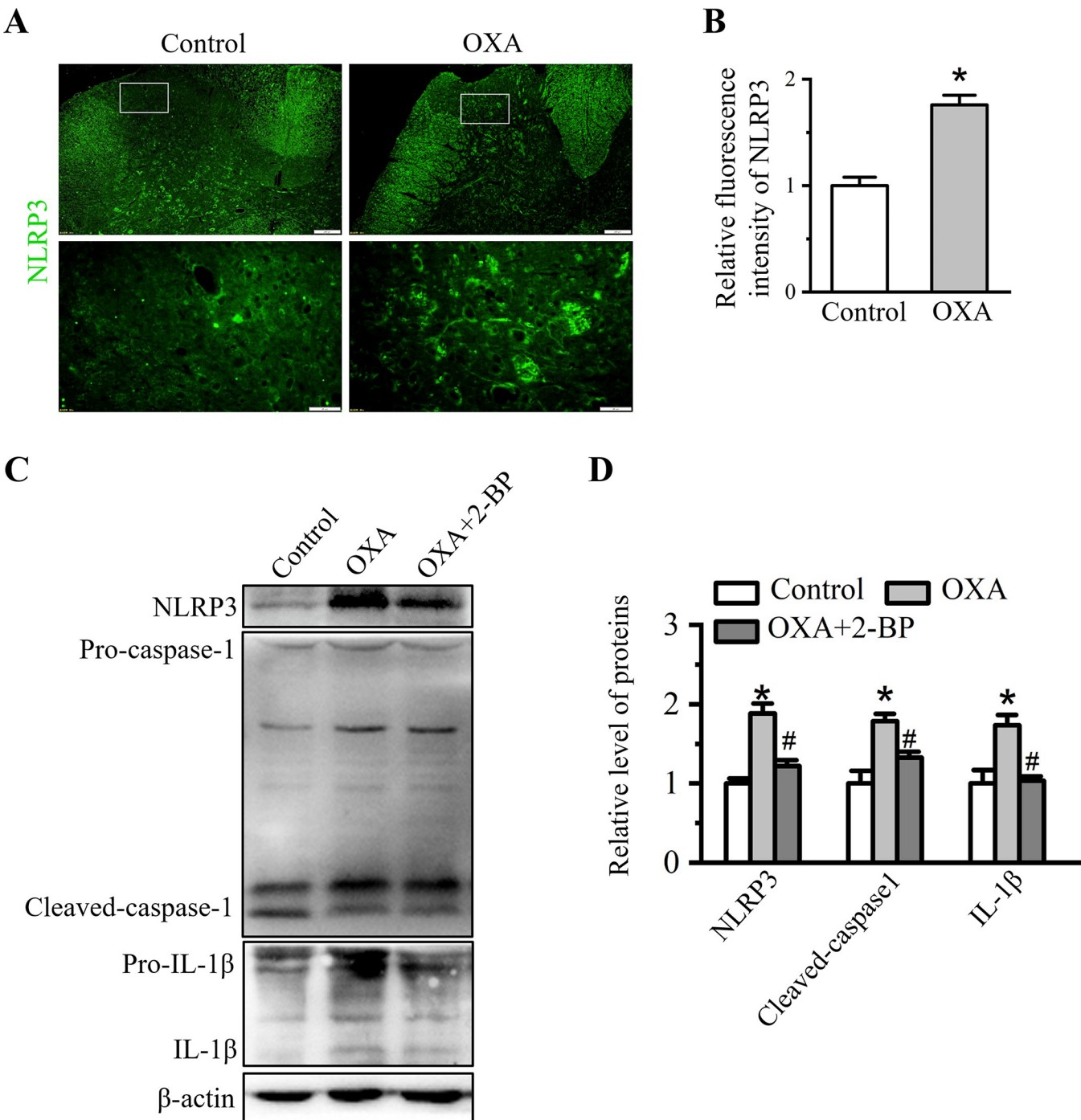

**Fig 3. Effect of 2-BP treatment on the components of NLRP3 inflammasome cascade.** (A) Representative immunofluorescence staining of spinal NLRP3 in Control and OXA groups. Scale bars: first line, 100 μm; second line, 20 μm. (B) Quantitative analysis of the fluorescence intensity of NLRP3. Data were showed as the mean ± SD (n = 3). *$P$<0.05 *vs*. Control group. (C) Western blot analysis of NLRP3, caspase-1 and IL-1β in Control, OXA and OXA+2-BP groups. (D) Quantitative analysis of protein levels in (C). Data were presented as mean ± SD (n = 3). *$P$ < 0.05 *vs*. Control group, #$P$ < 0.05 *vs*. OXA group.

were reduced to 1.22 ± 0.07, 1.32 ± 0.07, and 1.03 ± 0.06, respectively ($P$ < 0.05 *vs*. OXA group, Fig 3D). These data suggested that 2-BP treatment significantly reduced NLRP3 inflammasome mediated inflammatory response.

## 2-BP recovers mitochondrial function

In some neuropathology and neuroinflammation, mitochondrial dysfunction is accompanied by the release of pro-inflammatory cytokines. Herein, OXA treatment activated spinal inflammation have been demonstrated (Figs 1 and 2), we further detected whether OXA can affect mitochondrial function. Immunofluorescence staining and Western blot were used to examine the changes of spinal Drp1 expression and activity. Enhancement of fluorescence intensity of Drp1 in spinal cord of OXA-treated rats were observed (Fig 4A) and the relative intensity of Drp1 was 1.62 ± 0.09 ($P < 0.05$ *vs.* control group, Fig 4C). Drp1 GTPase was presented by the phosphorylation of Drp1 at Ser616. Our result showed that the phosphorylation of Drp1 at Ser616 was increased in spinal cord of OXA rats, and 2-BP treatment reduced this up-regulation (Fig 4F). Relative GTPase activity was represented by pDrp1/Drp1 and the relative activity of Drp1 in OXA group and OXA + 2-BP groups were 1.63 ± 0.07 ($P < 0.05$ *vs.* control group) and 0.79 ± 0.07 ($P < 0.05$ *vs.* OXA group, Fig 4G), respectively. The result showed that the up-regulation of phosphorylated Drp1 at Ser616 in spinal cord of OXA rats was reduced by 2-BP treatment.

Increased Drp1 GTPase activity initiated mitochondrial fission, mediated mitochondrial fragmentation, triggered COX2 release from mitochondria and promoted ROS production [34]. Excessive ROS production was scavenged by mitochondrial manganese superoxide dismutase (Mn-SOD). In spinal cord of OXA rats, the mean intensity of spinal COX2 was enhanced in OXA group (Fig 4B). Relative fluorescence intensity value of COX2 was 1.77 ± 0.09 ($P < 0.05$ *vs.* control group, Fig 4D). Western blot data showed that COX2 expression were increased to 1.42 ± 0.08 ($P < 0.05$ *vs.* control group) in OXA group (Fig 4F and 4G). Whereas Mn-SOD activity was dramatically decreased to 0.45 ± 0.05 ($P < 0.05$ *vs.* control group, Fig 4E) in OXA group. Intrathecal 2-BP administration reversed levels of COX2 to 1.09 ± 0.05 ($P < 0.05$ *vs.* OXA group, Fig 4G), and Mn-SOD activity to 0.75 ± 0.03 ($P < 0.05$ vs. OXA group, Fig 4E).

C6 cells were firstly treated with TNF-α to imitate inflammatory inducement, and then incubation of 2-BP to detect the changes in mitochondrial function. Mitochondrial membrane potential (MMP) represents mitochondrial activity which detected by Mito-Tracker Red assay. As shown in Fig 4H, TNF-α inducement resulted in significantly lower intensity which meant damaged MMP while 2-BP treatment recovered fluorescence intensity. Relative fluorescence intensity of Mito-Tracker Red in TNF-α and TNF-α+2-BP were 0.59 ± 0.11 ($P < 0.05$ *vs.* control group) and 0.88 ± 0.09 ($P < 0.05$ *vs.* TNF-α group, Fig 4I), respectively. Effect of 2-BP on ROS production was detected by the DCFH-DA indicator and decrease in DCFH-DA signal intensity was observed in 2-BP treated TNF-α-induced cells. As shown in Fig 4J, 2-BP decreased ROS fluorescence intensity upon TNF-α treatment. The relative ROS intensities of TNF-α and TNF-α + 2-BP groups were 3.28 ± 0.12 ($P < 0.05$ *vs.* control group) and 1.56 ± 0.16 ($P < 0.05$ *vs.* TNF-α group, Fig 4K), respectively.

These data indicated that 2-BP treatment significantly reduced Drp1 activity and recovered mitochondrial dysfunction.

## Discussion

OXA induced peripheral neuropathy is a well-recognized toxicity including acute and chronic neuropathy, resulting in increased pain sensation [35]. The current study used a neuropathic pain rat model induced by intraperitoneal injection of OXA. Dramatical increase in pain sensitivity were observed after 5 consecutive days intraperitoneal injection of OXA in our study which were coincident with research that rats showed a significant cold and mechanical allodynia upon OXA treatment [36, 37]. Neuronal components of the spinal cord dorsal horn

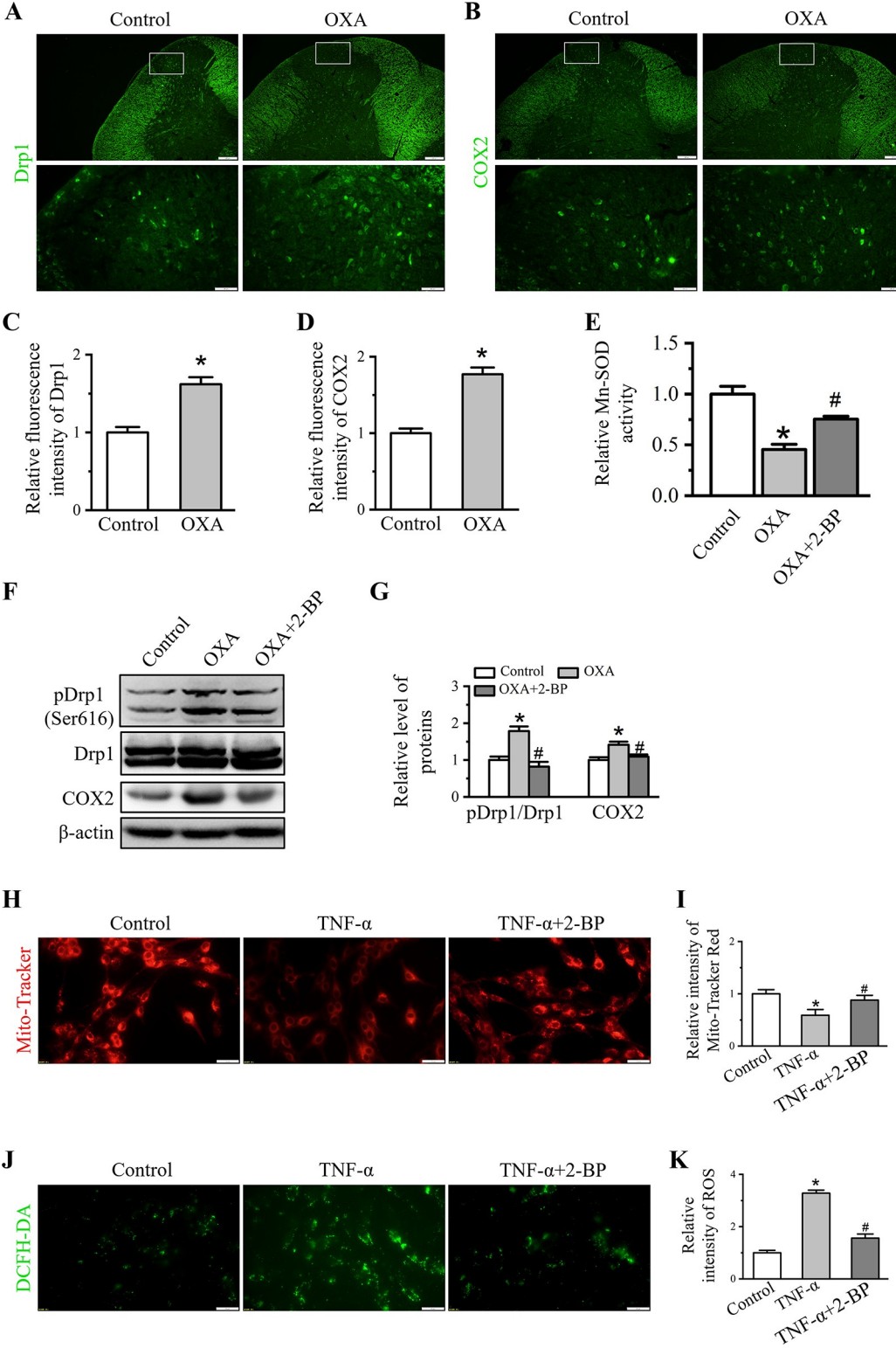

**Fig 4. Effect of 2-BP treatment on mitochondrial function.** (A, B) Representative immunofluorescence staining of spinal Drp1 (A) and COX2 (B) in control and OXA groups. Scale bars: first line, 100 μm; second line, 20 μm. (C-D) Quantitative analysis of the fluorescence intensity of Drp1 and COX2. Data showed as the mean ± SD (n = 3). (E) ELISA assay of spinal Mn-SOD activity of Control, OXA and OXA+2-BP groups. (F) Western blot analysis of pDrp1, Drp1 and COX2 in Control, OXA and OXA + 2-BP groups. (G) Western blot analysis of protein levels in (F). Data were presented as

mean ± SD (n = 3). $^*P < 0.05$ *vs.* Control group, $^#P < 0.05$ *vs.* OXA group. (H and J) Representative image showed change in MMP and ROS were detected in Control, TNF-α and TNF-α + 2-BP groups. Scale bars: 20 μm. (I and K) Quantitation of Mito-Tracker Red CMXRos (H) and DCFH-DA. Data were showed as the mean ± SD (n = 3). $^*P<0.05$ *vs.* Control group, $^#P<0.05$ *vs.* TNF-α group.

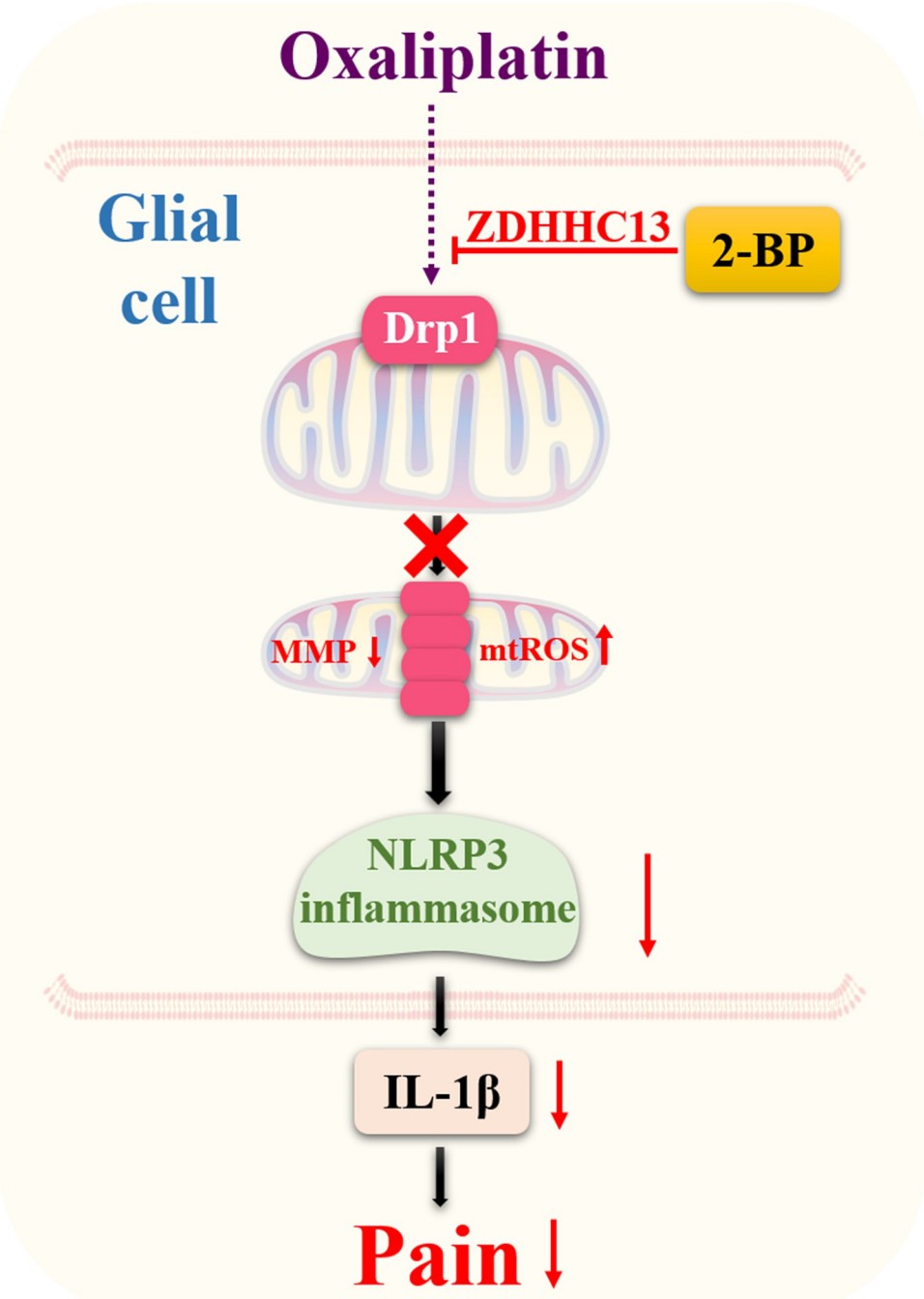

**Fig 5. Schematic representation of the potential mechanisms of 2-BP reduces spinal inflammation and ameliorates OXA-induced neuropathic pain.**

involved in the circuit participants in processing nociceptive information through receiving and integrating information from periphery and transmitting to brain [38]. Glia cells are the modulators of pain sensation. Astrocytes and microglia in spinal dorsal horn are activated in response to a wide array of conditions, including inflammation, bone cancer, and immune activation [39]. Pro-inflammatory cytokines mainly released by activated microglia and astrocytes. OXA could trigger spinal pathophysiological changes including the release of pro-inflammatory cytokines and the oxidative stress. Up-regulation of astrocytic marker GFAP and microglia marker Iba1 in spinal cord dorsal horn upon OXA treatment indicated that OXA induced spinal inflammatory response [40]. Under physiological conditions, glial cells keep in a resting state, whereas pathological conditions such as OXA induced neuropathy, they are activated. Microglia are in response to noxious signal within minutes while astrocytes activation usually occurs several days after the injury and much longer lasting. Activated microglia and astrocytes lead to release of pro-inflammatory cytokines which consequently cause neuronal damage [41, 42]. Our findings that activation of astrocytes and increased expression of IL-1β (Figs 2 and 3) were consistent with these research. Therefore, inhibition of microglia and astrocytes activation-mediated inflammatory cascade can be potential therapeutic targets for the treatment of chemotherapy-induced neuroinflammation.

Our previous research and published articles indicated that inflammation induced mitochondrial dysfunction is closely related to the activation of NLRP3 inflammasome [43, 44]. In the present study, factors essential to mitochondrial function and dynamics were dysregulated upon OXA administration. In OXA-treated rats, Drp1 and COX2 were increased while Mn-SOD activity was reduced which contributed to the activation of NLRP3 inflammasome cascade and consequent up-regulation of mature IL-1β was (Figs 1 and 2). In addition, mitochondria are facilitated to S-palmitoylation which mainly regulated by the protein acyltransferase [21]. ZDHHC13 is a protein acyltransferase and important for regulating mitochondrial dynamics. Deficiency of ZDHHC13 decreases Drp1 S-palmitoylation and consequently disrupted mitochondrial dynamics [22]. Herein, we found 2-BP inhibited the activity of ZDHHC13 decreased Drp1 GTPase activity, COX2 expression, NLRP3 inflammasome activation and ROS production induced by OXA treatment (Figs 1–3). All these changes accompanied by the increased PWT in OXA-induced rats upon 2-BP treatment. Taken together, 2-BP can decrease spinal inflammation and rescue mitochondrial function which consequently attenuates OXA-induced neuropathic pain.

## Conclusions

2-BP has an analgesic effect on chemotherapy-induced neuropathic pain. 2-BP reduces Drp1 GTPase activity through inhibiting protein acyltransferase activity of ZDHHC13. This action maintains mitochondrial dynamics and normal functions, and decreases NLRP3 cascade-mediated inflammatory reaction. The inhibitory effect of 2-BP on mitochondria dysfunction mediated inflammatory signal reverses OXA induced neuropathy pain (Fig 5).

## Supporting information

**S1 Checklist. The ARRIVE guidelines 2.0: Author checklist.**
(PDF)

**S1 File.**
(PDF)

## Author Contributions

**Conceptualization:** Min Xie.

**Data curation:** Zhi-Bin Dong, Min Xie.

**Formal analysis:** Min Xie.

**Funding acquisition:** Min Xie.

**Investigation:** Zhi-Bin Dong, Yu-Jia Wang, Bo-Jun Wang, Hong Lu, Ling Liu, Min Xie.

**Methodology:** Zhi-Bin Dong, Yu-Jia Wang, Meng-Lin Cheng, Bo-Jun Wang, Hong Lu, Hai-Li Zhu, Ling Liu, Min Xie.

**Project administration:** Bo-Jun Wang, Hai-Li Zhu, Ling Liu, Min Xie.

**Resources:** Yu-Jia Wang, Meng-Lin Cheng, Bo-Jun Wang, Hong Lu, Hai-Li Zhu, Ling Liu, Min Xie.

**Software:** Zhi-Bin Dong, Yu-Jia Wang, Meng-Lin Cheng, Bo-Jun Wang, Hong Lu, Hai-Li Zhu, Ling Liu, Min Xie.

**Supervision:** Yu-Jia Wang, Meng-Lin Cheng, Bo-Jun Wang, Hong Lu, Hai-Li Zhu, Ling Liu, Min Xie.

**Validation:** Zhi-Bin Dong, Yu-Jia Wang, Meng-Lin Cheng, Bo-Jun Wang, Hai-Li Zhu, Ling Liu, Min Xie.

**Visualization:** Zhi-Bin Dong, Yu-Jia Wang, Meng-Lin Cheng, Bo-Jun Wang, Hai-Li Zhu, Ling Liu, Min Xie.

**Writing – original draft:** Zhi-Bin Dong, Yu-Jia Wang, Meng-Lin Cheng, Ling Liu, Min Xie.

**Writing – review & editing:** Ling Liu, Min Xie.

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
