## [Decision Letter · Decision Letter 0]

1 Jun 2022

PONE-D-21-349902-Bromopalmitate decreases spinal inflammation and attenuates oxaliplatin-induced neuropathic pain via reducing Drp1-mediated mitochondrial dysfunctionPLOS ONE

Dear Dr. 

Thank you for submitting your manuscript to PLOS ONE. After careful consideration, we feel that it has merit but does not fully meet PLOS ONE’s publication criteria as it currently stands. Therefore, we invite you to submit a revised version of the manuscript that addresses the points raised during the review process.

We look forward to receiving your revised manuscript.

Kind regards,

Rosanna Di Paola, MD

Academic Editor

PLOS ONE

Journal Requirements:

2. As part of your revision, please complete and submit a copy of the Full ARRIVE 2.0 Guidelines checklist, a document that aims to improve experimental reporting and reproducibility of animal studies for purposes of post-publication data analysis and reproducibility: https://arriveguidelines.org/sites/arrive/files/Author%20Checklist%20-%20Full.pdf (PDF). Please include your completed checklist as a Supporting Information file. Note that if your paper is accepted for publication, this checklist will be published as part of your article.

4. Thank you for stating the following in the Funding Section of your manuscript:

“This study was supported by the grants from the National Natural Science Foundation of China (Nos. 81971066, 81901149 and 3210070630), Research Project of Hubei Provincial Department of Education (Nos. Q20192807, B2019167), Hubei University of Science and Technology Program (Nos. 2020TD02, BK20116).”

6. PLOS requires an ORCID iD for the corresponding author in Editorial Manager on papers submitted after December 6th, 2016. Please ensure that you have an ORCID iD and that it is validated in Editorial Manager. To do this, go to ‘Update my Information’ (in the upper left-hand corner of the main menu), and click on the Fetch/Validate link next to the ORCID field. This will take you to the ORCID site and allow you to create a new iD or authenticate a pre-existing iD in Editorial Manager. Please see the following video for instructions on linking an ORCID iD to your Editorial Manager account: https://www.youtube.com/watch?v=_xcclfuvtxQ.

7. PLOS ONE now requires that authors provide the original uncropped and unadjusted images underlying all blot or gel results reported in a submission’s figures or Supporting Information files. This policy and the journal’s other requirements for blot/gel reporting and figure preparation are described in detail at https://journals.plos.org/plosone/s/figures#loc-blot-and-gel-reporting-requirements and https://journals.plos.org/plosone/s/figures#loc-preparing-figures-from-image-files. When you submit your revised manuscript, please ensure that your figures adhere fully to these guidelines and provide the original underlying images for all blot or gel data reported in your submission. See the following link for instructions on providing the original image data: https://journals.plos.org/plosone/s/figures#loc-original-images-for-blots-and-gel

8. We note that you have included the phrase “data not shown” in your manuscript. Unfortunately, this does not meet our data sharing requirements. PLOS does not permit references to inaccessible data. We require that authors provide all relevant data within the paper, Supporting Information files, or in an acceptable, public repository. Please add a citation to support this phrase or upload the data that corresponds with these findings to a stable repository (such as Figshare or Dryad) and provide and URLs, DOIs, or accession numbers that may be used to access these data. Or, if the data are not a core part of the research being presented in your study, we ask that you remove the phrase that refers to these data.

9. Your ethics statement should only appear in the Methods section of your manuscript. If your ethics statement is written in any section besides the Methods, please move it to the Methods section and delete it from any other section. Please ensure that your ethics statement is included in your manuscript, as the ethics statement entered into the online submission form will not be published alongside your manuscript.

Reviewers' comments:

Reviewer's Responses to Questions

**Comments to the Author**

1. Is the manuscript technically sound, and do the data support the conclusions?

Reviewer #1: Yes

Reviewer #2: Yes

2. Has the statistical analysis been performed appropriately and rigorously? 

Reviewer #1: Yes

Reviewer #2: Yes

3. Have the authors made all data underlying the findings in their manuscript fully available?

Reviewer #1: Yes

Reviewer #2: Yes

4. Is the manuscript presented in an intelligible fashion and written in standard English?

Reviewer #1: Yes

Reviewer #2: Yes

5. Review Comments to the Author

Reviewer #1: The manuscript must describe a technically sound piece of scientific research with data that supports the conclusions. the authors made all data underlying the findings described in their manuscript fully available without restriction. The manuscript is presented in an intelligible fashion and written in standard English.

Reviewer #2: Manuscript: 2-Bromopalmitate decreases spinal inflammation and attenuates oxaliplatin-induced neuropathic pain via reducing Drp1-mediated mitochondrial dysfunction

Authors: Zhi-Bin Dong, Yu-Jia Wang , Meng-Lin Cheng , Bo-Jun Wang, Hong Lu , Hai-Li Zhu , Ling Liu, Min Xie.

ID: PONE-D-21-34990

Overall recommendation:

Accept after addressing minor revisions

General comments:

The paper by Zhi-Bin Dong and colleagues add another important element regarding the toxicity of platinum derivate Oxaliplatin and on the use of 2-Bromopalmitate to counteract these effects.This manuscript represents a good starting point as reference data for future studies on this topic. Despite the present study has a good experimental idea, it needs revisions both for contents and structural organization in order to improve the quality of the manuscript. Some acronyms are missing, written in full, eg: DCFH-DA and PWT, please add them in the text.

Keywords:

It would be advisable to change the keywords; some of these are also present in the paper's title.

Background:

It would be appropriate to include more information about oxaliplatin toxicity, so I suggest that the authors include the following reference on zebrafish model in which many aspects related to oxaliplatin toxicity from both morphological and molecular points of view are emphasized. DOI: 10.3390/toxics10020081.

Materials and Methods;

Materials and Methods section needs revision:

It would be opportune in indicate the full addresses of the manufacturers from whom the materials used for conducting the experiments are procured, please standardize the whole section accordingly.

It would be desirable to improve the quality of the description of the methods used as well,I suggest taking a cue for the exposition of the Western Blot procedure from the following manuscript and insert the appropriate citation : DOI: 10.3390/life12010128

The authors need to implement the quality of even the purely technical sections ,for SOD measurement take a cue and cite: DOI: 10.3390/toxins13100710

Same for ROS assessment section, take a cue from the text and cite it: DOI: 10.3390/toxics9120344

The authors stated: “After TNF-α inducement and 2-BP treatment, C6 cells were loaded with MitoTracker Red CMXRos or DCFH-DA and incubated at 37 ℃ for 30 min”.It’s not clear what the author used for this assay, was MitoTracker Red CMXRos or DCFH-DA?,please revise.

Figures: I suggest the authors to arrange the legend figures ,sometimes there are letters corresponding to a photo in brackets,sometimes not,it becomes very difficult to follow what the authors are trying to describe

6. PLOS authors have the option to publish the peer review history of their article (what does this mean?). If published, this will include your full peer review and any attached files.

Reviewer #1: No

Reviewer #2: No

---

## [Author Response · Author response to Decision Letter 0]

9 Jul 2022

Journal Requirements:

Response: We have modified the article format according to the PLOSOne formatting sample title authors affiliations and PLOSOne formatting sample main body templates in the revised manuscript.

2. As part of your revision, please complete and submit a copy of the Full ARRIVE 2.0 Guidelines checklist, a document that aims to improve experimental reporting and reproducibility of animal studies for purposes of post-publication data analysis and reproducibility: 

https://arriveguidelines.org/sites/arrive/files/Author%20Checklist%20-%20Full.pdf (PDF). Please include your completed checklist as a Supporting Information file. Note that if your paper is accepted for publication, this checklist will be published as part of your article.

Response: We completed the Checklist and uploaded as a Supporting Information file.

Response: We have checked and corrected the Funding Information in the revised manuscript.

4. Thank you for stating the following in the Funding Section of your manuscript:

“This study was supported by the grants from the National Natural Science Foundation of China (Nos. 81971066, 81901149 and 3210070630), Research Project of Hubei Provincial Department of Education (Nos. Q20192807, B2019167), Hubei University of Science and Technology Program (Nos. 2020TD02, BK20116).”

Response: We have modified the Funding Information according to the PLOSOne formatting sample main body templates and deleted the Acknowledgements in the revised manuscript.

We also have added the “The funders had no role in study design, data collection and analysis, decision to publish, or preparation of the manuscript” to the Funding part in the revised manuscript.

The amended statements were added in the cover letter.

Response: The Data Availability Statement in the revised manuscript was amended as follows: All relevant data are within the paper.

6. PLOS requires an ORCID iD for the corresponding author in Editorial Manager on papers submitted after December 6th, 2016. Please ensure that you have an ORCID iD and that it is validated in Editorial Manager. To do this, go to ‘Update my Information’ (in the upper left-hand corner of the main menu), and click on the Fetch/Validate link next to the ORCID field. This will take you to the ORCID site and allow you to create a new iD or authenticate a pre-existing iD in Editorial Manager. Please see the following video for instructions on linking an ORCID iD to your Editorial Manager account: https://www.youtube.com/watch?v=_xcclfuvtxQ.

Response: The corresponding author has a validated ORCID iD.

7. PLOS ONE now requires that authors provide the original uncropped and unadjusted images underlying all blot or gel results reported in a submission’s figures or Supporting Information files. This policy and the journal’s other requirements for blot/gel reporting and figure preparation are described in detail at https://journals.plos.org/plosone/s/figures#loc-blot-and-gel-reporting-requirements and https://journals.plos.org/plosone/s/figures#loc-preparing-figures-from-image-files. When you submit your revised manuscript, please ensure that your figures adhere fully to these guidelines and provide the original underlying images for all blot or gel data reported in your submission. See the following link for instructions on providing the original image data: https://journals.plos.org/plosone/s/figures#loc-original-images-for-blots-and-gel

Response: The original uncropped and unadjusted images underlying all blot or gel results were uploaded.

8. We note that you have included the phrase “data not shown” in your manuscript. Unfortunately, this does not meet our data sharing requirements. PLOS does not permit references to inaccessible data. We require that authors provide all relevant data within the paper, Supporting Information files, or in an acceptable, public repository. Please add a citation to support this phrase or upload the data that corresponds with these findings to a stable repository (such as Figshare or Dryad) and provide and URLs, DOIs, or accession numbers that may be used to access these data. Or, if the data are not a core part of the research being presented in your study, we ask that you remove the phrase that refers to these data.

Response: The phrase “data not shown” was deleted in the revised manuscript.

9. Your ethics statement should only appear in the Methods section of your manuscript. If your ethics statement is written in any section besides the Methods, please move it to the Methods section and delete it from any other section. Please ensure that your ethics statement is included in your manuscript, as the ethics statement entered into the online submission form will not be published alongside your manuscript.

Response: The ethics statement was added in the Methods section in the revised manuscript. 

Response: We checked the reference and corrected the wrongly cited literature and cited some latest literatures. 

Reviewers' comments:

General comments:

The paper by Zhi-Bin Dong and colleagues add another important element regarding the toxicity of platinum derivate Oxaliplatin and on the use of 2-Bromopalmitate to counteract these effects. This manuscript represents a good starting point as reference data for future studies on this topic. Despite the present study has a good experimental idea, it needs revisions both for contents and structural organization in order to improve the quality of the manuscript. Some acronyms are missing, written in full, eg: DCFH-DA and PWT, please add them in the text.

Response: Both the contents and structural organization has been modified and the full written of the abbreviations has been added in the revised manuscript. In addition, the keywords have been adjusted. 

Keywords:

It would be advisable to change the keywords; some of these are also present in the paper's title.

Response: The keywords in the revised manuscript were “chemotherapy pain, oxaliplatin, 2-bromopalmitate, NLRP3 inflammasome, dynamin-related protein 1 (Drp1), manganese superoxide dismutase (Mn-SOD), cyclooxygenase-2 (COX-2)”

Background:

It would be appropriate to include more information about oxaliplatin toxicity, so I suggest that the authors include the following reference on zebrafish model in which many aspects related to oxaliplatin toxicity from both morphological and molecular points of view are emphasized. DOI: 10.3390/toxics10020081.

Response: The oxaliplatin toxicity on the changes in the tissues was added in the Introduction section of the revised manuscript as follows: “As an anti-cancer agent, OXA induced histopathological changes in the heart, liver, intestines and muscle in a dose dependent manner”.

Materials and Methods;

Materials and Methods section needs revision:

It would be opportune in indicate the full addresses of the manufacturers from whom the materials used for conducting the experiments are procured, please standardize the whole section accordingly.

Response: The full address of the manufacturers and catalogue number of the antibodies and kits were added in the Antibodies and reagents section. 

It would be desirable to improve the quality of the description of the methods used as well, I suggest taking a cue for the exposition of the Western Blot procedure from the following manuscript and insert the appropriate citation : DOI: 10.3390/life12010128

Response: The details of Western Blot procedure were added according to the reference (DOI: 10.3390/life12010128) in the Western blot analysis section.

The authors need to implement the quality of even the purely technical sections, for SOD measurement take a cue and cite: DOI: 10.3390/toxins13100710

Same for ROS assessment section, take a cue from the text and cite it: DOI: 10.3390/toxics9120344

Response：The details of SOD and ROS measurements were added and the references were cited in the Assessment sections of revised manuscript. 

The authors stated: “After TNF-α inducement and 2-BP treatment, C6 cells were loaded with MitoTracker Red CMXRos or DCFH-DA and incubated at 37 ℃ for 30 min”.It’s not clear what the author used for this assay, was MitoTracker Red CMXRos or DCFH-DA?,please revise.

Response: The mitochondrial membrane potential measurement and ROS measurement have been described respectively as two separated sections and the details of the processes were added in the revised manuscript.

Figures: I suggest the authors to arrange the legend figures ,sometimes there are letters corresponding to a photo in brackets,sometimes not,it becomes very difficult to follow what the authors are trying to describe.

Response: The letters represented the figures of photo have been added in the descriptions.

---

## [Decision Letter · Decision Letter 1]

19 Sep 2022

2-Bromopalmitate decreases spinal inflammation and attenuates oxaliplatin-induced neuropathic pain via reducing Drp1-mediated mitochondrial dysfunction

PONE-D-21-34990R1

Dear Dr. xie,

We’re pleased to inform you that your manuscript has been judged scientifically suitable for publication and will be formally accepted for publication once it meets all outstanding technical requirements.

Kind regards,

Rodrigo Franco

Academic Editor

PLOS ONE

Additional Editor Comments (optional):

Reviewers' comments:

Reviewer's Responses to Questions

**Comments to the Author**

1. If the authors have adequately addressed your comments raised in a previous round of review and you feel that this manuscript is now acceptable for publication, you may indicate that here to bypass the “Comments to the Author” section, enter your conflict of interest statement in the “Confidential to Editor” section, and submit your "Accept" recommendation.

Reviewer #1: All comments have been addressed

Reviewer #2: All comments have been addressed

2. Is the manuscript technically sound, and do the data support the conclusions?

Reviewer #1: Yes

Reviewer #2: Yes

3. Has the statistical analysis been performed appropriately and rigorously? 

Reviewer #1: Yes

Reviewer #2: Yes

4. Have the authors made all data underlying the findings in their manuscript fully available?

Reviewer #1: Yes

Reviewer #2: Yes

5. Is the manuscript presented in an intelligible fashion and written in standard English?

Reviewer #1: Yes

Reviewer #2: Yes

6. Review Comments to the Author

Reviewer #1: Thanks for addressing all comments and corrections of the manuscript which represents a very good and informative research in this field

Reviewer #2: The authors answered all my comments and improved the quality of the manuscript which is now suitable for publication.

7. PLOS authors have the option to publish the peer review history of their article (what does this mean?). If published, this will include your full peer review and any attached files.

Reviewer #1: No

Reviewer #2: No

---

## [Editor Report · Acceptance letter]

20 Oct 2022

PONE-D-21-34990R1 

2-Bromopalmitate decreases spinal inflammation and attenuates oxaliplatin-induced neuropathic pain via reducing Drp1-mediated mitochondrial dysfunction 

Dear Dr. xie:

I'm pleased to inform you that your manuscript has been deemed suitable for publication in PLOS ONE. Congratulations! Your manuscript is now with our production department. 

Kind regards, 

on behalf of

Dr. Rodrigo Franco 

Academic Editor

PLOS ONE